# Attention-based Multiple Instance Learning with Mixed Supervision on the Camelyon16 Dataset

**Paul Tourniaire**                                              PAUL.TOURNIAIRE@INRIA.FR
*Universite Cote d'Azur, Inria, Epione Team, Sophia Antipolis, France*

**Marius Ilie**                                                      ILIE.M@CHU-NICE.FR
**Paul Hofman**                                                  HOFMAN.P@CHU-NICE.FR
*Laboratory of Clinical and Experimental Pathology, Hospital-Integrated Biobank (BB-0033-00025),
Nice Hospital University, FHU OncoAge, Universite Cote d'Azur, 06000 Nice, France*

**Nicholas Ayache**                                          NICHOLAS.AYACHE@INRIA.FR
**Hervé Delingette**                                      HERVE.DELINGETTE@INRIA.FR
*Universite Cote d'Azur, Inria, Epione Team, Sophia Antipolis, France*

**Editor:** TBA

## Abstract

Since the standardization of Whole Slide Images (WSIs) digitization, the use of deep learning methods for the analysis of histological images has shown much potential. However, the sheer size of WSIs is a real challenge, as they are often up to 100,000 pixels wide and high at the highest resolution, and therefore cannot be processed directly by any model. Moreover, as the manual delineation of structures within WSIs is tedious, histological datasets often only contain slide-level labels, or a limited amount of delineated slides. In this context, multiple-instance learning (MIL) approaches have been proposed, considering WSIs as bags of smaller images, designated as tiles or patches. Among these methods, the attention-based MIL aims at learning the importance of each tile for the slide final classification while at the same time performing a clustering of those tiles. In this paper, we introduce the concept of mixed supervision within this framework, by exploiting tile-level labels in addition to slide-level labels to improve the classification of slides. More precisely, we show on the Camelyon16 dataset that even a small proportion of slides with pixel-wise annotations can improve their classification but also the localization of tumorous regions. This improves the consistency of the results between the tile and slide levels and the interpretability of the algorithm.

**Keywords:** Attention Mechanism, Mixed Supervision, Histopathology

## 1. Introduction

In terms of tumor assessment, histopathology is currently the clinical gold-standard diagnosis technique. Nonetheless, pathologists face multiple challenges when confronted to Whole Slide Images (WSIs), and often require careful and time-intensive efforts. WSIs are usually stained with Hematoxylin and Eosin (H&E) and scrutinized through a microscope by the pathologist in order to detect cancerous tissue. Given the size of tissue samples, and the potential artifacts that may occur, such as tissue folds or tears, the diagnosis is often

tedious. Since the digitization of pathological slides, machine and deep learning algorithms have offered automated solutions for the diagnosis of tumors (Bejnordi et al., 2017; Wang et al., 2019). However, pixel-level annotations such as tumor segmentation are usually unavailable, since the annotation process requires both time and medical expertise: datasets often only display slide-level labels. Among deep learning algorithms for weakly-supervised learning, some of the most popular are based on the multiple instance learning assumption (MIL). As such, the WSI is divided into tiles or patches of smaller size (e.g. 512 x 512 pixels$^2$) and the slide is considered normal if all instances (tiles) are normal, and tumorous if at least one tile contains tumor. The tile-level predictions are then aggregated following various mechanisms to provide the slide-level or patient-level label (Campanella et al., 2019; Courtiol et al., 2018).

The attention mechanism was recently proposed as the tile pooling function (Ilse et al., 2018). The authors propose to extract tile-level features with a convolutional network (Sirinukunwattana et al., 2016), and use a two-layered neural network to calculate a weighted average and select key instances for the slide-level prediction. A recent improvement of the attention-based MIL was proposed in the CLAM (Clustering-constrained Attention Multiple instance learning) algorithm (Lu et al., 2021): the introduction of instance-level clustering during training, where instances given the highest (resp. lowest) attention scores were considered positive (resp. negative) evidence of the slide class. A smooth SVM loss (Berrada et al., 2018) calculated on the instance-level prediction using an instance classifier is then added to the overall slide-level cross-entropy. For the tile-level feature extraction, a frozen ImageNet pre-trained ResNet50 (He et al., 2016) is used, so as to accelerate the training procedure. This, however, was identified as a major caveat by Dehaene et al. (2020), who decided to replace the backbone with a self-supervised deep neural network, trained on histological images using contrastive loss (He et al., 2020). On the Camelyon16 dataset (Bejnordi et al., 2017), they showed performance close to the best performing fully-supervised method (Wang et al., 2016), but at the cost of heavy and long GPU training.

When pixel-level labels are available, the combined use of pixel-level and image-level labels is coined *mixed supervision*. It is especially useful when the amount of pixel-level annotated images is low, in regards to the total amount of available images. Mlynarski et al. (2019) showed that using weakly and fully annotated data to train a deep learning model for brain tumor segmentation improved the segmentation performance compared to the baseline trained only on fully annotated data. Ciga and Martel (2021) simultaneously trained a ResNet18 model on classification and segmentation tasks, leveraging only a part of available fully annotated data, to obtain the same performance as models trained on the entire fully labeled dataset. However, both classification and segmentation tasks relied on tile-level information, the authors having devised a strategy to select tiles suited for either task.

In this work, starting from the CLAM framework, we propose the first mixed supervision approach within the attention-based MIL framework. Our main goal is not only to improve the slide-level classification, but also the localization performance of the model. Our contributions are listed as follows:

- First, we introduce mixed supervision for the MIL framework inside the CLAM algorithm.

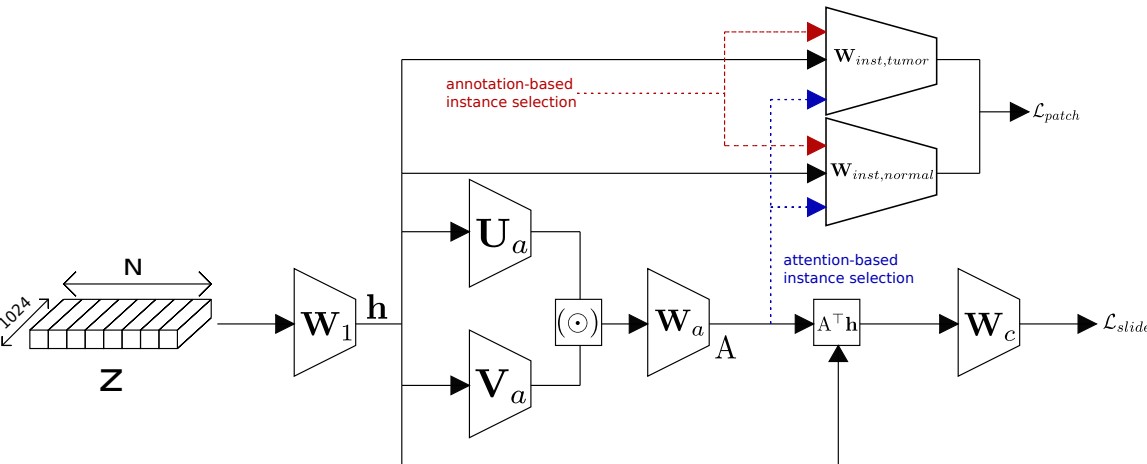

Figure 1: Overview of the CLAM model. Activation functions are not detailed in the interest of clarity. The original instance selection approach appears in blue (based on the attention scores), whereas our annotation-based instance selection approach appears in red.

- Second, we propose an improvement of the instance-level clustering method in the case of non-pathological slides.

- Finally, we show that we can improve the model's classification performance and the localization of tumorous tiles by adding a small amount of ground-truth tile-level labels along with slide-level labels.

## 2. Methods

### 2.1 Data preprocessing

For the preprocessing of the WSIs, we first transform a downsampled version of the image to the HSV space and apply Otsu thresholding (Otsu, 1979) on the hue and saturation channels to detect tissue. Then, we extract 256 x 256 non-overlapping tiles from the tissue region, at the highest magnification level. For the tiles feature extraction, we use the same model as Lu et al. (2021), i.e. a ResNet50 architecture without the final classification layer, and an output dimension of 1024.

### 2.2 CLAM Algorithm

#### 2.2.1 DESCRIPTION OF THE ORIGINAL MODEL

**Slide-level classification**. The overview of the model can be seen in Figure 1. We consider a classification problem with $n$ classes, and detail only the "multiple branches" version of CLAM (CLAM MB). A slide is represented as a feature matrix $\mathbf{z}$, of size 1024 x N, where N is the number of tiles in the slide. A first fully-connected layer $\mathbf{W}_1$ transforms each 1024-dimensional feature vector $\mathbf{z}_k$ into a 512-dimensional feature vector $\mathbf{h}_k$, where $k$ is the tile index. Then a gated-attention module computes the attention weights from $\mathbf{h}$ for each

of the $n$ classes, following:

$$a_{k,m} = \frac{\mathbf{W}_{a,m}(\tanh(\mathbf{V}_a\mathbf{h}_k^\top) \odot \mathrm{sigm}(\mathbf{U}_a\mathbf{h}_k^\top))}{\sum_{i=1}^{N} \exp\{\mathbf{W}_{a,m}(\tanh(\mathbf{V}_a\mathbf{h}_i^\top) \odot \mathrm{sigm}(\mathbf{U}_a\mathbf{h}_i^\top))\}} \tag{1}$$

where $m$ is the class index, and $\mathbf{V}_a$, $\mathbf{U}_a$, and $\mathbf{W}_a$ are fully-connected layers. The attention weights are represented by the matrix $\mathrm{A} \in \mathbb{R}^{\mathrm{N} \times n}$. Before the final binary classification layers $\mathbf{W}_{c,m}$, each feature vector $\mathbf{h}_k$ is multiplied by its corresponding attention weight. This operation is represented by the matrix $\mathrm{M} = \mathrm{A}^\top\mathbf{h}$. Finally, each column of M (representing one of the $n$ classes) is independently processed by one of the $n$ classifiers to obtain the slide-level score for each class. A softmax activation function is eventually applied on the logits, and a cross-entropy loss $\mathcal{L}_{slide}$ is computed on the obtained scores.

**Instance-level clustering**. Based on the attention scores $a_{k,m}$, the tiles with the highest (resp. lowest) scores are retrieved for each class, and assigned a positive (resp. negative) pseudo-label. The assumption is that high-attention tiles are positive evidences of the class, whereas the low-attention tiles are negative evidences. For each class, a binary classifier $\mathbf{W}_{inst,m}$ classifies the gathered instances, and computes a tile-level smooth top-1 SVM loss $\mathcal{L}_{patch}$ which is added to $\mathcal{L}_{slide}$ to give the global loss term: $\mathcal{L} = c_1\mathcal{L}_{slide} + c_2\mathcal{L}_{patch}$ where $c_1$ and $c_2$ are hyperparameters.

The instance-level clustering does not directly affect the slide-level classification, rather, it encourages the learning of discriminative features by the model layer $\mathbf{W}_1$ to better separate classes. Note that the inference of slide-level classification does not use the instance level clustering.

### 2.2.2 Instance-level clustering for non-pathological slides

Given a tumor diagnosis task on a histopathological dataset, where some slides contain tumors, and others not, there is a caveat with the current formulation of the instance-level clustering: indeed, when assigning negative pseudo-labels to low-attention tiles, the model is actually learning that these tiles are *not* normal, and should therefore be classified as such. However, it is one of the hypotheses of MIL to consider all instances of a non-tumorous slide to be tumor-free. Therefore, all tiles originating from a non-tumorous slide should be classified as non-tumorous too. As a result, we decide to ignore the instance classifier $\mathbf{W}_{inst,normal}$, and instead use the tumor instance classifier $\mathbf{W}_{inst,tumor}$ to classify the $B$ patches with the highest attention scores in normal slides as negative evidence of tumors. That way, we expect to reduce the number of false positives in the final classification. Furthermore, we also ignore $\mathbf{W}_{inst,normal}$ to classify the patches from tumorous slides, and use only $\mathbf{W}_{inst,tumor}$. Indeed, without the need for histological subtyping, only a single binary classifier is required to perform the distinction between tumorous and non tumorous images. $\mathbf{W}_{inst,normal}$ remains therefore unused in our model.

### 2.2.3 Mixed supervision: Instance-level classification with ground-truth labels

Instead of selecting the tiles based on the highest and lowest attention scores, we introduce a mixed supervision formulation of the instance-level clustering which is now qualified as *instance-level classification*. On slides where tumorous regions were delineated by expert

annotators, we propose to select tiles based on whether they belong to those tumorous regions. In doing so, we are sure that the instance-level classification layer learns with instances truly representative of two classes which is not the case when they are selected based on their attention score. This selection is only performed on slides for which annotations are available, which may be a small proportion of the histological training set. In fact, we hope to improve the model performance in classification and tumor localization (i.e. the tile-level classification of the slide) even with a small ratio of tile-level labels.

The training stage is therefore divided into two parts: first, only the subset of slides with pixel-wise annotations along with a subset of normal slides is used to train the model. This is to train the instance level classifier on true classes without the noise generated by the original pseudo-labeling procedure. We use various subset proportions (10, 50, 80% of the training slides) to measure the impact of the annotations on the results. For a tumorous slide containing N tiles such that $N = N_{tumor} + N_{normal}$, we randomly sample $min(K, N_{tumor})$ tiles from the tumorous ones, and $min(K, N_{normal})$ among the normal ones, where K is a hyperparameter. Then, positive and negative labels are generated for the two sets of tiles, and a score is assigned to each tile $t$ following: $l_t = \mathbf{W}_{inst,tumor}\mathbf{h}_t$. Second, the entire training set is used, this time without using any instance-level label, leveraging only slide-level labels. This procedure is summarized by Algorithm 1.

### 2.2.4 Implementation details

During training, we used a batch size of 1. We used the Adam optimizer (Kingma and Ba, 2014) with $\beta_1 = 0.9$ and $\beta_2 = 0.99$, and a learning rate of $2.10^{-4}$. All models were trained on a single NVIDIA GeForce GTX 1080 GPU. We used 70 epochs with early stopping after 20 epochs without improvement on the slide-level cross-entropy loss in the validation set.

During the first part of the training, we set $c_1 = c_2 = 0.5$ (when using tile-level labels) so that both loss terms $\mathcal{L}_{slide}$ and $\mathcal{L}_{patch}$ are weighted equally, and $c_1 = 0.7$ and $c_2 = 0.3$ during the second part, for these were the values used by (Lu et al., 2021) in their article. For hyperparameter K, we tested several increasing values (128, 256, 512, 1024, 5000), and we used the value K = 1024, for which we reached the best performance. The value $B = 8$ was kept from the original model.

## 3. Results

### 3.1 Data description and experiments

#### 3.1.1 The Camelyon16 dataset

The Camelyon16 challenge (Bejnordi et al., 2017) aimed at detecting metastases in H&E-stained WSIs of lymph node sections. The dataset contains 399 slides, split between a training set of 270 slides, and a test set of 129 slides. The slides were prepared and stained in two different medical centers. All slides that contain metastases (111 slides in the training set, 49 in the test set) have been exhaustively annotated (except for 20 of them in the training set, partially) by a group of expert pathologists. Annotations are available as XML files and can be converted to binary masks using the Automated Slide Analysis Platform (ASAP) open source software (`https://github.com/computationalpathologygroup/ASAP`). All slides were scanned at 40x magnification ($\simeq 0.25\mu m$/pixel).

---

**Algorithm 1:** Instance-level classification using tile-level labels

---

**Data:** $(\mathbf{h}_1, ..., \mathbf{h}_N)$, $Y$ (the slide label), K, $B$

**Result:** $l_Y$

$l_Y \leftarrow \{\}$

**if** $Y = $ *"tumor"* **then**

$\quad$ K$_{tumor} = min($K, N$_{tumor})$

$\quad$ K$_{normal} = min($K, N$_{normal})$

$\quad$ Select $t_1, ..., t_{\mathrm{K}_{tumor}}$ $\qquad\qquad\qquad\qquad$ `// tumor tiles indexes`

$\quad$ Select $t'_1, ..., t'_{\mathrm{K}_{normal}}$ $\qquad\qquad\qquad\qquad$ `// normal tiles indexes`

$\quad$ **for** $t \leftarrow t_1, ..., t_{\mathrm{K}_{tumor}}$ **do**

$\quad\quad$ generate positive label $y_t = 1$

$\quad\quad$ $l_t = \mathbf{W}_{inst,tumor}\mathbf{h}_t$

$\quad$ **end**

$\quad$ $l_Y \leftarrow l_Y \cup \{l_t\}$

$\quad$ **for** $t' \leftarrow t'_1, ..., t'_{\mathrm{K}_{normal}}$ **do**

$\quad\quad$ generate negative label $y_{t'} = 0$

$\quad\quad$ $l'_t = \mathbf{W}_{inst,tumor}\mathbf{h}'_t$

$\quad$ **end**

$\quad$ $l_Y \leftarrow l_Y \cup \{l'_t\}$

**else**

$\quad$ Select $t_1, ..., t_B$

$\quad$ **for** $t \leftarrow t_1, ..., t_B$ **do**

$\quad\quad$ generate negative label $y_t = 0$

$\quad\quad$ $l_t = \mathbf{W}_{inst,tumor}\mathbf{h}_t$

$\quad$ **end**

$\quad$ $l_Y \leftarrow l_Y \cup \{l_t\}$

**end**

---

### 3.1.2 EXPERIMENTAL SETUP

To measure the classification performance of our models, we decided to use the area under the receiver operating characteristic curve (AUC), as it was the metric used for the challenge, along with the F1-score. All models were evaluated on the challenge test set. The training set was further split into a training and a validation set. We used 5-fold cross validation to perform these splits in order to select the best performing model. In the training set, we randomly sampled $k\%$ ($k \in \{10, 50, 80\}$) of the slides to use with tile-level labels outside of the 20 with only partial annotations.

For the localization performance, we computed slide binary masks based on the outputs of the instance-clustering layer $\mathbf{W}_{inst,tumor}$ after having applied a threshold of 0.5. The masks were computed at the $5^{\text{th}}$ resolution level. Furthermore, we used two different metrics for normal and tumorous slides: in the former, we computed the tile-wise specificity, i.e. the amount of tiles erroneously classified as tumor divided by the total amount of tiles. In the latter, we computed the Dice score in reference to the ground-truth mask. This metric was computed on 46 slides from the test set, as 3 metastatic slides out of the 49 were unavailable

| Method | AUC | F1-score | Mean Dice score (std) | Mean tile-level Specificity (std) |
|---|---|---|---|---|
| CLAM | 0.895 | 0.8 | 0.215(0.28) | 0.864(0.1) |
| CLAM w/ 10% annot. | 0.924 | 0.835 | 0.35(0.263) | 0.999(0.001) |
| CLAM w/ 50% annot. | 0.939 | **0.878** | 0.375(0.279) | **0.999(0.001)** |
| CLAM w/ 80% annot. | **0.949** | 0.873 | **0.405(0.282)** | 0.999(0.002) |

Table 1: Classification and localization metrics for the different methods.

(one because the annotation file is absent from the dataset, the other two because of an error when computing the reference masks from the annotation file in ASAP).

## 3.2 Evaluation

The test set classification results are shown in Table 1. For all models, we report the performance from the best fold, as it is usually done in challenges. We can see that the models with tile-level labels all outperform the CLAM algorithm in terms of slide-level classification, even when only 10% of the annotated slides were used. Moreover, we can see that the models trained with tile-level labels have also higher scores on localization tasks, both for tumorous (mean dice scores) and normal slides (mean specificity). In particular, models trained with tile-level labels tend to detect less falsely tumorous tiles than the reference model. Figure 2 shows an example of a tumorous slide from the test set and the masks computed using CLAM and two other annotation-based models. We can see that although the tumorous region is quite well detected in CLAM, there are many false positive tiles. When using tile-level labels however, these false positives tend to disappear, and the detected region is closer to the ground truth. Figure 3 is another example from the test set of the difference between CLAM and annotation-based models. Figure 4 shows an example of a normal slide (also taken from the test set): here again, we can see that there are many more false positives when using CLAM. Using only 10% (resp. 80%) of the annotated slides, our method would have ranked 6th (resp. 4th) on the challenge open leaderboard.

## 4. Discussion & Conclusion

In this work, we presented a mixed supervision approach for attention-based MIL. We proposed to add strong supervision in the tile classification branch of the CLAM algorithm for a subset of the training slides. We showed that even with a small amount (10%) of slides with pixel-wise annotations of tumors, we were able to obtain improved classification of slides and above all a better localization of the tiles with tumour tissue. In particular, we witnessed a sharp decrease in the number of false positive tiles, i.e. a better discrimination between tiles with normal and tumorous tissue. This better localization of tumors is crucial for WSI with only slide-level labels, since machine learning algorithms may provide the correct classification at the slide level but not necessarily with correct classifications at the tile level. This work shows that even with fairly limited pixel-wise annotation, it is possible to obtain more consistent and robust results at both local (tile) and global (slide) levels. Finally, the better localization of tumorous tiles and reduced rate of false positives in

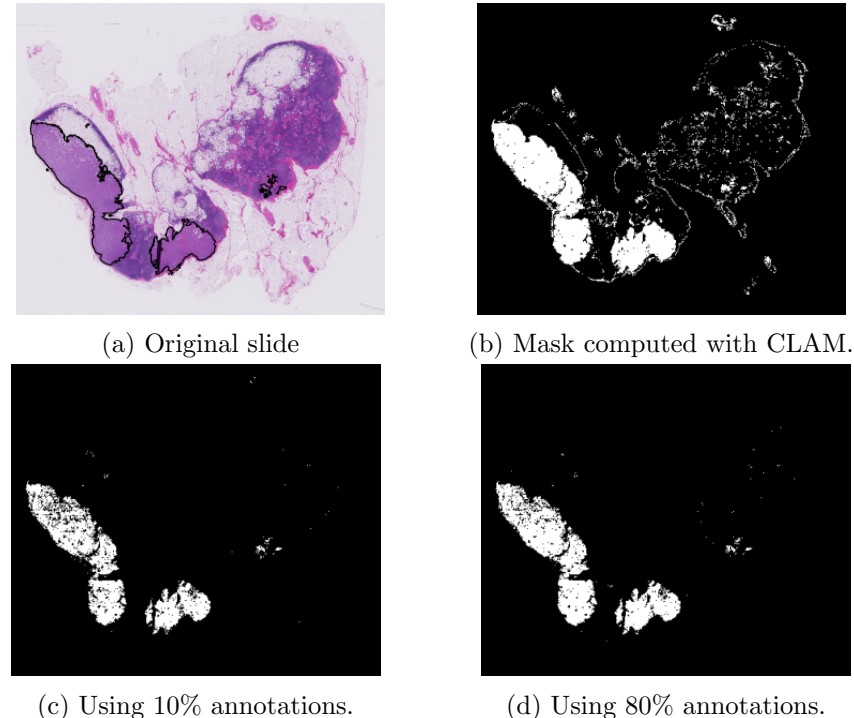

(a) Original slide        (b) Mask computed with CLAM.

(c) Using 10% annotations.        (d) Using 80% annotations.

Figure 2: Metastatic slide *test_016* from Camelyon16 (the metastasis region is delineated in black), next to binary masks computed using the different models. (b) displays the results of the CLAM algorithm, (c) and (d) show the results obtained using 10% and 80% of tile-level labels.

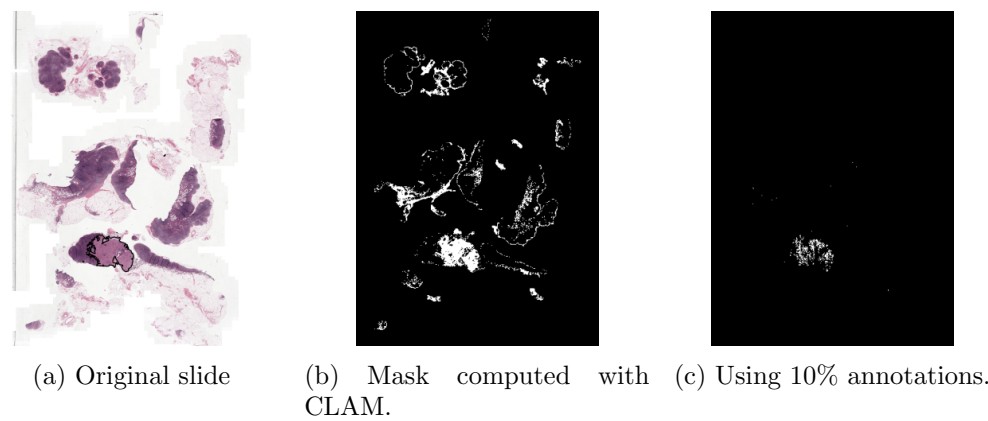

(a) Original slide     (b) Mask computed with CLAM.     (c) Using 10% annotations.

Figure 3: Metastatic slide *test_068* from Camelyon16 (the metastasis region is delineated in black), next to binary masks computed using CLAM, and the model with 10% of tile-level labels.

non-pathological slides also improves the interpretability of the provided slide classification algorithm which is key for the adoption of those algorithms in clinical practice.

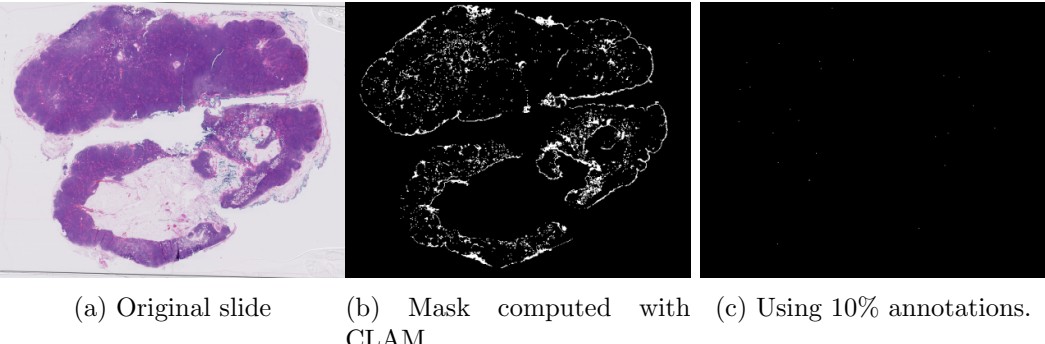

(a) Original slide  (b) Mask computed with  (c) Using 10% annotations.
CLAM.

Figure 4: Slide *test_042* from Camelyon16 without tumor, next to binary masks computed using CLAM, and the model with 10% of tile-level labels.

The proposed approach may be improved in several ways. We have noticed that on the original CLAM and on our mixed supervised version, the localization performance of tumorous tiles may vary from one fold to the next. For our method, one or two folds (out of five) have significantly worse results than the other folds. This may be due to the lack of stain normalization performed since the slides originate from two distinct centers with different acquisition equipment. Considering that only two centers were involved in the acquisition of the challenge data, we originally did not consider that stain normalization was essential. Moreover, in the method that won the challenge (Wang et al., 2016), some post-processing steps were applied to improve the localization accuracy. In our case, no post-processing was performed, and only a coarse localization map was computed, which could be refined for improved accuracy. Finally, the original CLAM paper considered the attention weights $a_{k,m}$ rather than the instance-level clustering as the source of information to localize pathologies. Therefore, it may also be interesting to supervise the attention mechanism with pixel-wise annotation similarly to what we proposed in this paper on the tile classification.

## Acknowledgments

The authors are grateful to the OPAL infrastructure from Université Côte d'Azur for providing resources and support. This work has been supported by the French government, through the 3IA Côte d'Azur Investments in the Future project managed by the National Research Agency (ANR) with the reference number ANR-19-P3IA-0002

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
