# OpenReview forum: "Attention-based Multiple Instance Learning with Mixed Supervision on the Camelyon16 Dataset"
_MICCAI.org/2021/Workshop/COMPAY — COMPAY 2021_

### Official Review · Reviewer_NgHv · 2021-08-03
**The paper introduces a mixed supervision approach to attention-based MIL, that improves the diagnosis of cancerous WSI, as well as the localization of tumor regions within the WSI.**

**Rating:** 8
**Confidence:** 3

**Review:**

**Strengths**

The paper contributions are clearly listed and are significant. The paper is coherent and well written and includes a comprehensive literature study. The results convincingly show that their method improves classification performance and reduces the number of false positive tumor tiles.


**Weaknesses**

A rather small dataset (Camelyon16) is used: 270 training and 129 test WSI. More details of the steps of the model is needed.


**Questions to address**

1.	More detailed description is needed to explain how the tile-level predictions from the instance-level classification are used in the second step of the slide-level prediction.
2.	Clarify that the examples in figures 2, 3 and 4 are from the held-out test set. Examples from the test set should be presented.
3.	You mention in the discussion that some folds from the cross-validation perform much worse, possibly due to lack of stain normalization. Motivate why you did not normalize the slides in the pre-processing?
4.	How did you decide hyperparameters? Present all hyperparameter values used. E.g. what was the minimum number of tiles used (K), global loss term weighting (c1 and c2) etc?


**Minor comments**

1.	Define “CLAM” on first use.
2.	Define “ROC” on first use.
3.	“…we report the performance from the best fold,...” should be clarified to “best fold model” or such, not to be confused with performance on the fold data.

---

### Official Review · Reviewer_er36 · 2021-08-17
**Attention-based Multiple Instance Learning with Mixed Supervision on the Camelyon16 Dataset**

**Rating:** 6
**Confidence:** 5

**Review:**

In this paper, the tile level annotations are used along with slide level labels to train an attention-based MIL approach.

The paper is mainly based on "CLAM" and "attention-based MIL" by Maximilian Ilse, where they have an extra branch to add tile-level supervision. The way that tile-level labels are used is interesting, and it is shown that this extra supervision can improve performance. However,  the paper needs some clarification:
1- Is the model first trained on slides with instance-level labels and then is fine-tuned on all slides without using instance-level labels? Because the training procedure is divided into two parts, it seems it happens at two different stages? if yes, why the two parts are not unified.

2- The process of using W(inst,normal) and W(inst, tumor) is not clear from the paper. During inference which one should be selected for the final tile level predictions! How these two are combined, is either of them used during training?

3- It is also good to see if the clustering itself is effective or not. I mean utilizing only labels as supervision. Considering only instance-level supervision for slides with the pixel-wise labels, and ignore the rest of slides. It is not clear if clustering slides to high attention and low attention and then assigning labels to them is really effective or not.

4- What is the performance using 100% annotations? It will show the maximum capacity of the model compared to other models in Camelyon challenge leaderboard. And if comparable performance to the 1st ranked method can be achieved, it can be shown that post-processing steps that were used by other methods (extracting features, random forest, etc.) may not be required.

---

### Decision · Program_Chairs · 2021-08-25

Accept